# Short-term responses of temperate and subarctic marine diatoms to Irgarol 1051 and UV radiation: Insights into temperature interactions

Dongquan Bi[1☯], Lixin Cao[1☯], Yuheng An[1], Juntian Xu[1,2], Yaping Wu[1,2]*

**1** Jiangsu Key Laboratory of Marine Bioresources and Environment/Co-Innovation Center of Jiangsu Marine Bio-Industry Technology, Jiangsu Ocean University, Lianyungang, China, **2** Key Laboratory of Coastal Salt Marsh Ecosystems and Resources Ministry of Natural Resources, Jiangsu Ocean University, Lianyungang, China

☯ These authors contributed equally to this work.
* ypwu@jou.edu.cn

**Data Availability Statement:** All relevant data are within the paper and its Supporting Information files.

## Abstract

Phytoplankton face numerous pressures resulting from chemical and physical stressors, primarily induced by human activities. This study focuses on investigating the interactive effects of widely used antifouling agent Irgarol 1051 and UV radiation on the photo-physiology of marine diatoms from diverse latitudes, within the context of global warming. Our findings clearly shown that both Irgarol and UV radiation have a significant inhibitory impact on the photochemical performance of the three diatoms examined, with Irgarol treatment exhibiting more pronounced effects. In the case of the two temperate zone diatoms, we observed a decrease in the inhibition induced by Irgarol 1051 and UVR as the temperature increased up to 25˚C. Similarly, for the subarctic species, an increase in temperature resulted in a reduction in the inhibition caused by Irgarol and UVR. These results suggest that elevated temperatures can mitigate the short-term inhibitory effects of both Irgarol and UVR on diatoms. Furthermore, our data indicate that increased temperature could significantly interact with UVR or Irgarol for temperate diatoms, while this was not the case for cold water diatoms, indicating temperate and subarctic diatoms may respond differently under global warming.

## Introduction

Phytoplankton, serving as the fundamental basis of marine ecosystems, possess remarkable adaptability to cope with challenging environmental conditions [1]. However, under the ongoing climate change scenario, the gradual increase in global sea surface temperature, coupled with the frequent occurrence of heatwaves in recent years, poses additional challenges for these microorganisms, that might attributed to the global decline of phytoplankton biomass [2,3]. Temperature has a significant impact on almost all metabolic pathways, with the intensity and duration of temperature exposure playing crucial roles [4], and its interactions with

**Funding:** This research was funded by the Natural Science Foundation of China (41876113) to Y. Wu, the Key Project of Natural Science of Jiangsu High School (21KJA170001) to Y. Wu, Qinglan Project of Jiangsu Province to Y. Wu, and Priority Academic Program Development of Jiangsu Higher Education Institutions to Y. Wu. The funders had no role in the study design, data collection and analysis, decision to publish, or preparation of the manuscript.

**Competing interests:** The authors have declared that no competing interests exist.

other environmental stressors, such as UV radiation, nutrients, and organic pollutants, have also been well documented [5,6].

Ultraviolet radiation (UVR) is a crucial environmental factor, significantly influencing the phytoplankton inhabiting in the euphotic zone [7]. UVR can constrain enzyme activity, consequently diminishing photosynthesis and growth rates [8], while also can cause DNA damage, potentially leading to mutations and ultimately cell death of phytoplankton [9]. However, positive effects were observed under some conditions [10]. The increasing levels of UVR due to climate change is becoming a major concern for marine ecosystems [11]. With the receding sea ice and shoaling of upper mixed layer, phytoplankton are subjected to increasing levels of UVR, which can negatively affect their growth and productivity [12]. Such impacts could initiate a cascading effect on the marine food web, considering that phytoplankton are an essential component of marine ecosystems [13].

Anthropogenic activities released significant amounts of hazardous waste containing a wide range of organic compounds into the ocean, leading to detrimental effects on the photosynthesis and metabolic pathways of phytoplankton [14]. With the rapid expansion of international trade and the shipping industry, booster biocides have been detected globally, including in high latitude regions [15]. One of the major pollutant is Irgarol 1051, a common triazine compound, which poses environmental risks in areas with high shipping traffic even at relatively low concentrations [16]. Its presence has been linked to the potential alteration of community structure, as it can selectively favor tolerant taxa [17,18]. Additionally, there is evidence suggesting that this pollutant may interact synergistically with UVR, further impacting the physiology of phytoplankton [19].

Photosystem II (PSII) is a complex of proteins and pigments that plays a crucial role in the process of photosynthesis [20]. It is responsible for converting the energy from captured photons by pigments into electron flow, which in turn generates chemical energy needed for carbon dioxide fixation [21]. Compared to Photosystem I (PSI), PSII is more vulnerable to environmental stressors, UVR and high-intensity photosynthetically active radiation (PAR) can cause damage to PSII, disrupting the electron transport chain and leading to the formation of reactive oxygen species [22]. UVR targets the subunits of photosystem, and the pollutant Irgarol can also bind to these subunits, impeding electron transfer and directly promoting photodegradation of PSII [23]. To maintain efficient photosynthetic performance, damaged PSII subunits need to be replaced by newly synthesized proteins through a series of enzymatic reactions [24], while this process is sensitive to temperature changes [6]. Previous studies have shown that elevated temperatures counteracted the negative effects of UVR on the growth and photosynthetic efficiency of eukaryotic phytoplankton, as well as cyanobacteria [25,26], as well as mitigate the acute toxicity of the commonly used herbicide diuron to green algae [27].

Given the complexed environment faced by phytoplankton, we proposed a hypothesis that temperature might influence the interplay between Irgarol 1051 and UVR, impacting the photosynthetic performance of marine phytoplankton. To explore this hypothesis, we focused on three marine diatoms from different latitudes and examined their responses to Irgarol 1051, UVR, and temperature, which are considered as representative environmental factors in this study.

## Materials and methods

### Species, chemicals and culture conditions

In present work, we selected three diatoms that maintained in laboratory for over 2 years, namely two temperate centric diatoms, *Thalassiosira weissflogii* (CCMA102) and *Skeletonema costatum* (JOUP006) that were isolated from the Daya Bay (N22°42′10″, E114°39′36″) and

Yellow Sea (N34°41′24″, E119°30′15″), respectively, that were grown at 20°C, and a subarctic pennate diatom *Entomoneis* sp. (JOUP008) that was originally isolated from the Bering Strait (N64°30′15″, E190°18′20″), and grown at 5°C. Extensive studies have been conducted on the response of two temperate diatoms to environmental factors, while few study focused on the subarctic species [11,28]. The light condition was around 50 μmol photons m$^{-2}$ s$^{-1}$ for the culture stock, to keep a relatively slow growth rate. The diatoms used in the present study were obtained from the laboratory culture stock rather than a field site, so no permit was required.

The diatoms were inoculated into sterilized natural seawater that was enriched to F/2 medium [29], and subjected to semi-continuous culturing in triplicate 1 L polycarbonate bottles for a duration of 10 days within an outdoor water bath. To mimic natural light condition, the polycarbonate bottles were exposed to solar radiation, which allowed for UVA transmission while blocking almost all UVB radiation. All bottles were shaded by a neutral density mesh, which had approximately 25% transmission. The mean daily light intensity during the experiment was around 120 μmol photons m$^{-2}$ s$^{-1}$. To ensure the availability of nutrients, approximately 60% of the culture volume was exchanged with fresh medium every two days, thus the cells to be maintained at the exponential phase, and the cell density was kept at an optimal level. During the daytime, the bottles were manually shaken 4 to 5 times to mimic natural turbulence. Following the shaking, the bottles were randomly placed back into the water bath to allow for further experiments under the specified conditions.

Irgarol 1051 (2-methylthio-4-tert-butylamino-6-cyclopropylamino-striazine) was prepared in DMSO (C$_2$H$_6$OS), at a concentration of 0.4 mg L$^{-1}$ for the stock solution. Preliminary experiments confirmed that the DMSO had no effect on the photochemical performance of the marine diatoms across a range of temperatures from 5 to 25°C.

## Experimental setup

For the temperate diatoms, the temperature was controlled using a thermostat, with three different temperature settings: 10, 15, or 20°C, which represented the temperature range typically observed in temperate oceans, and a temperature of 5°C was maintained for the subarctic species. In order to simulate the global warming scenario, a 5°C temperature increase was applied for three diatoms, the variation for each temperature was controlled to vary by less than 1°C (Fig 1).

After a minimum acclimation period of 10 days at various temperatures, sub-cultures of the grown cells were carefully collected and dispensed into quartz tubes with a volume of 100 mL. These tubes were then subjected to specific conditions for a duration of 120 minutes, the exposure light intensity was set to ~120 μmol photons m$^{-2}$ s$^{-1}$ for PAR, and 4.2 W m$^{-2}$ for UVR. Previous studies have shown that the photo-physiology of diatoms would reach a stable state after 60 min under stressful condition [19]. To ensure proper adaptation to the experimental conditions, each sample was allowed to acclimate for 10 minutes prior to commencing the experiment. The quartz tubes were then placed into a water bath at 30-second intervals, ensuring that each tube experienced the same illumination conditions throughout the experiment. These illumination conditions were achieved using a fluorescence lamp, which emitted PAR, and a Q-Panel lamp (UVA-340, Miami, USA), which emitted both UVA and UVB radiation to generate UVR. To create specific treatments, cut-off glass filters (ZJB280 or ZJB400) were placed on top of the quartz tubes, that will block the radiation below wavelengths of 280 nm or 400 nm, and produced PAR+UVR (PAB) and PAR treatments. These filters selectively allowed the desired wavelengths of light to pass through, ensuring that the appropriate radiation combination was achieved within the tubes. Furthermore, the compounds Irgarol 1051 or DMSO (as a control) were added into the quartz tubes, thus 4 treatments were created as shown in Table 1.

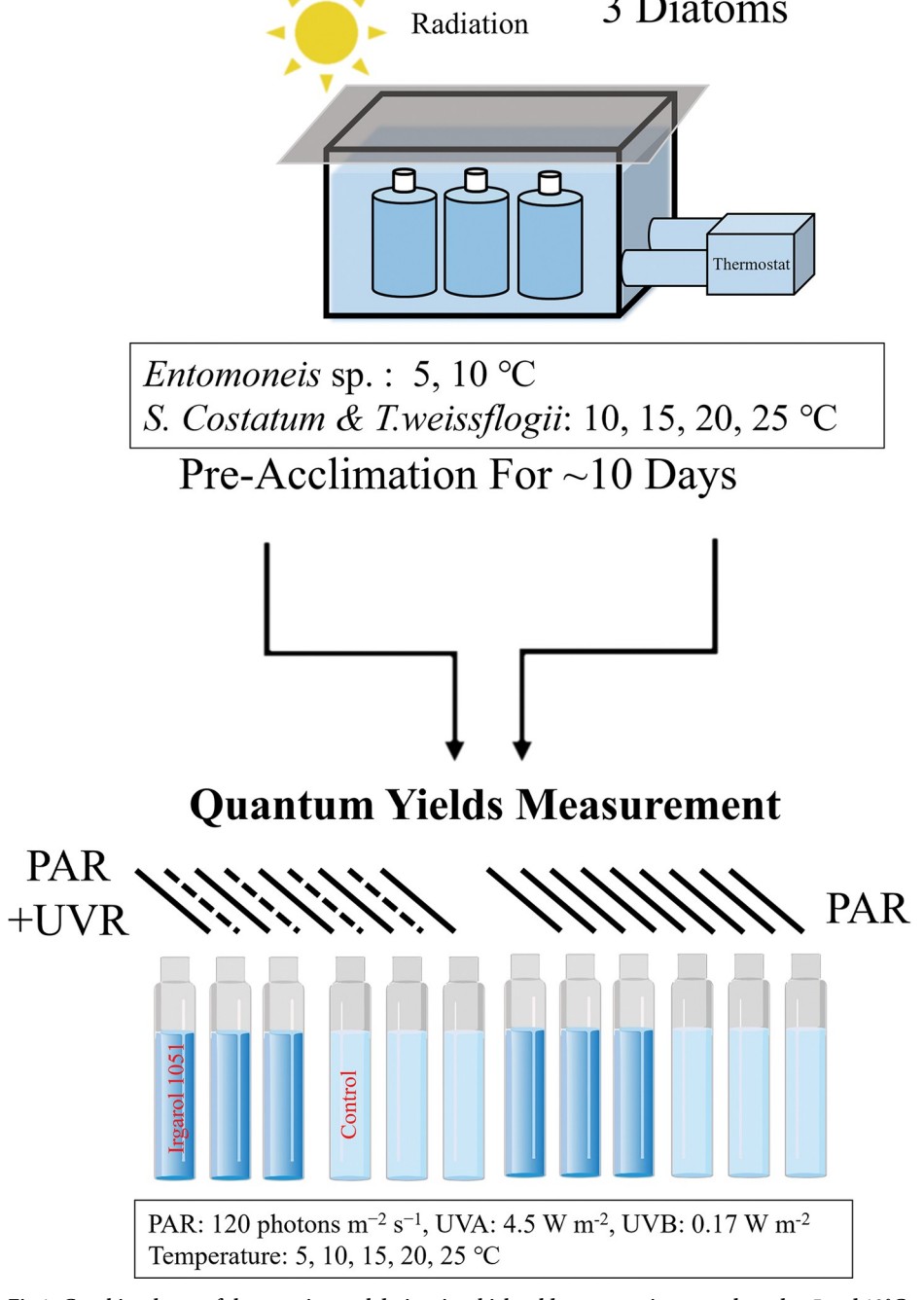

**Fig 1. Graphic scheme of the experimental design, in which cold water species was cultured at 5 and 10°C, temperate species were at 10, 15, 20 and 25°C.** After pre-acclimation at the different temperatures for 10 days, cultures were dispensed into quartz tubes, with addition of 0.1% DMSO as control, and 0.4 μg L$^{-1}$ Irgarol 1051 (dissolved in DMSO), then placed under customized equipment for the 120 min exposure experiment (see text for details).

The nominal Irgarol concentration in the sub-cultures was 0.4 μg L$^{-1}$, which is closed to reported environmental levels, particularly in coastal waters [16]. To ensure consistent light exposure conditions, the distance between the lamps and the quartz tubes was adjusted

**Table 1. Treatment matrix for exposure experiments.**

| Factors | DMSO | DMSO + Irgarol |
|---|---|---|
| PAR | Treatment 1 (T1) | Treatment 3 (T3) |
| PAR + UVR | Treatment 2 (T2) | Treatment 4 (T4) |

T1: PAR control

T2: PAB control

T3: PAR + Irgarol and

T4: PAB + Irgarol.

accordingly, and the accuracy of the light intensity was validated using a portable radiometer (PMA2100, Solar Light, USA).

## Determination of chlorophyll fluorescence

For chlorophyll fluorescence measurement, quartz tubes with cells were incubated under PAR or PAB conditions. Then sub-sample (~2 mL) was taken from each tube regularly and measured within 10 seconds by an Aquapen fluorometer (Photon Systems Instruments, Czech Republic) with a saturating light pulse of 3000 μmol photons $m^{-2}$ $s^{-1}$. This procedure without dark adaptation allowed us to get the real photochemical performance at respective conditions. The time interval between measurements was 6 min for the first 30 min and 10–15 min for the rest of measurements.

## Data analysis

Quantum yields were calculated from the fluorescence values measured with the Aquapen fluorometer.

Effective quantum yield (QY) = $(F_m' - F_t)/F_m'$

Where $F_m'$ represents the maximal fluorescence, and $F_t$ represents the steady-state fluorescence under actinic light [30].

The relative inhibition induced by UVR, Irgarol and UVR + Irgarol (U + I) on PSII was calculated as following:

Relative UVR inhibition (%) = $(Y_{control} - Y_{PAB})/ Y_{control} \times 100\%$

Relative Irgarol inhibition (%) = $(Y_{control} - Y_{Irgarol})/ Y_{control} \times 100\%$

Relative U + I inhibition (%) = $(Y_{control} - Y_{PAB + Irgarol})/ Y_{control} \times 100\%$

Under stressful condition in algae, photosystem II activity decreases until the damage and repair of PSII reach a balance, thereafter remaining at a quasi-steady state. In the present study this occurred after 60 min exposure, thus $Y_{control}$, $Y_{PAB}$, $Y_{Irgarol}$ and $Y_{PAB + Irgarol}$ represent the averaged quantum yields from 60 to 120 min exposure under respective treatments.

The individual and interactive effects of temperature, UVR, and Irgarol on diatoms were analyzed using a permutation 3-way ANOVA. Before conducting the analysis, the assumptions of homoscedasticity for each data group and normality of residuals were verified. To determine the significance of differences in relative inhibition among treatments, a one-way ANOVA with Tukey's test was performed.

To evaluate the directions of two-level interactions between UVR, Irgarol, or temperature, we calculated the expected quantum yield ($QY_e$) assuming an additive interaction between these factors, and compared this with the measured quantum yield ($QY_m$) obtained under the corresponding treatment.

If the ratio $QY_m/QY_e$ is greater than 1, it indicates an antagonistic interaction, a ratio of 1 suggests an additive interaction, while a ratio less than 1 implies a synergistic interaction [31].

## Results

The QY of *T. weissflogii* under PAR remained stable throughout the exposure experiment, maintaining values around 0.55–0.59 (Fig 2). However, under UVR and/or Irgarol treatment, the QY gradually decreased and eventually reached stable levels towards the end of the exposure. At a temperature of 10°C, the QY was reduced to 0.33 under UVR alone and further decreased to 0.26 under Irgarol treatment (Fig 2A), the lowest QY was observed when exposed to both UVR and Irgarol, reaching around 0.19 at the end of the exposure period (Fig 2A). At a temperature of 15°C, the QY decreased to 0.51 under UVR alone and dropped to 0.35 under Irgarol treatment, while the lowest QY value was also recorded under the combined treatment of UVR and Irgarol, approximately 0.29 (Fig 2B). Similar patterns were observed at temperatures of 20°C and 25°C, where the QY exhibited a gradual decrease as seen at 10°C and 15°C, although the magnitude of the decrease was generally smaller (Fig 2C and 2D).

The trend in QY for *S. costatum* at a temperature of 10°C showed a similar pattern to that observed for *T. weissflogii* (Fig 3A). The lowest QY value, approximately 0.32, was recorded

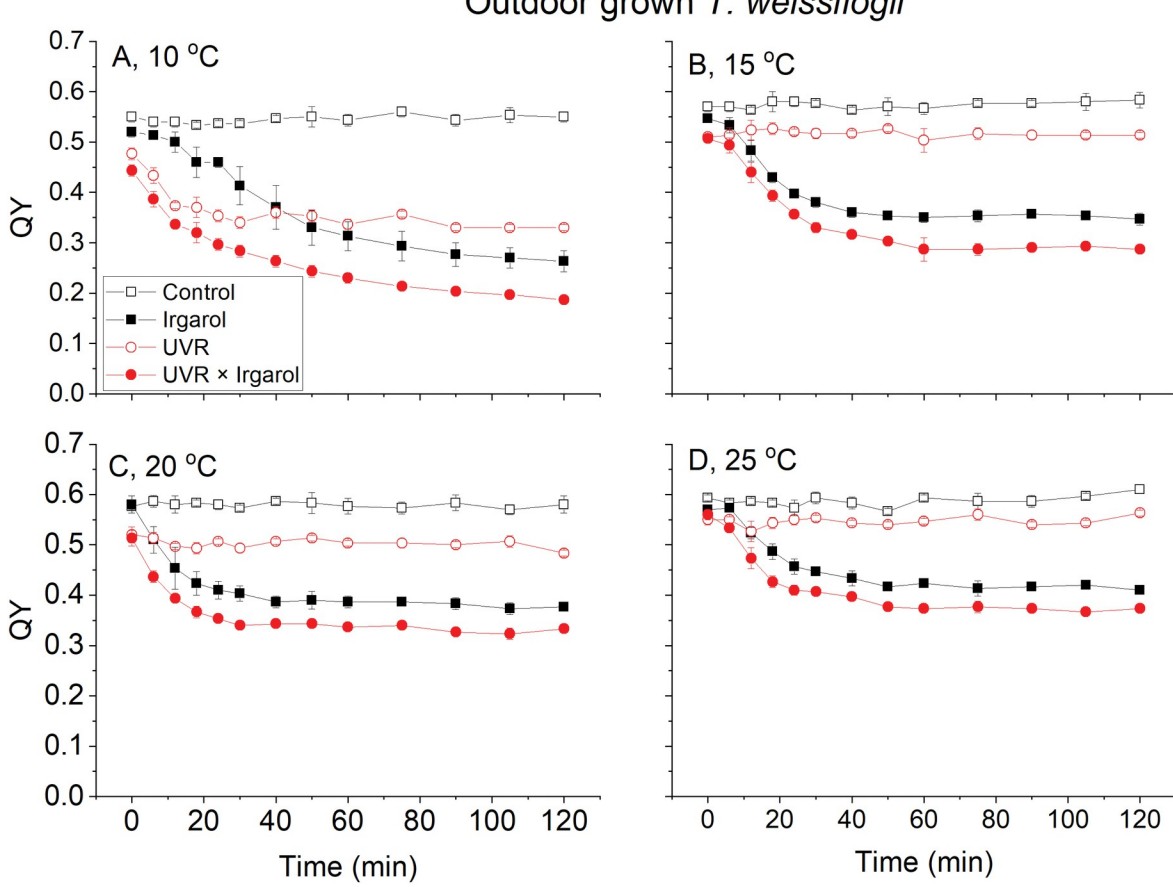

**Fig 2.** The quantum yields (QY) of photosystem II of *T. weissflogii* at four temperatures (A: 10°C, B: 15°C, C: 20°C, D: 25°C) and exposed to two radiation treatments (PAR, black and UVR, red) and two Irgarol 1051 treatments (0, hollow symbols and 0.4 μg L$^{-1}$, solid symbols). Vertical lines represent standard deviations, n = 3.

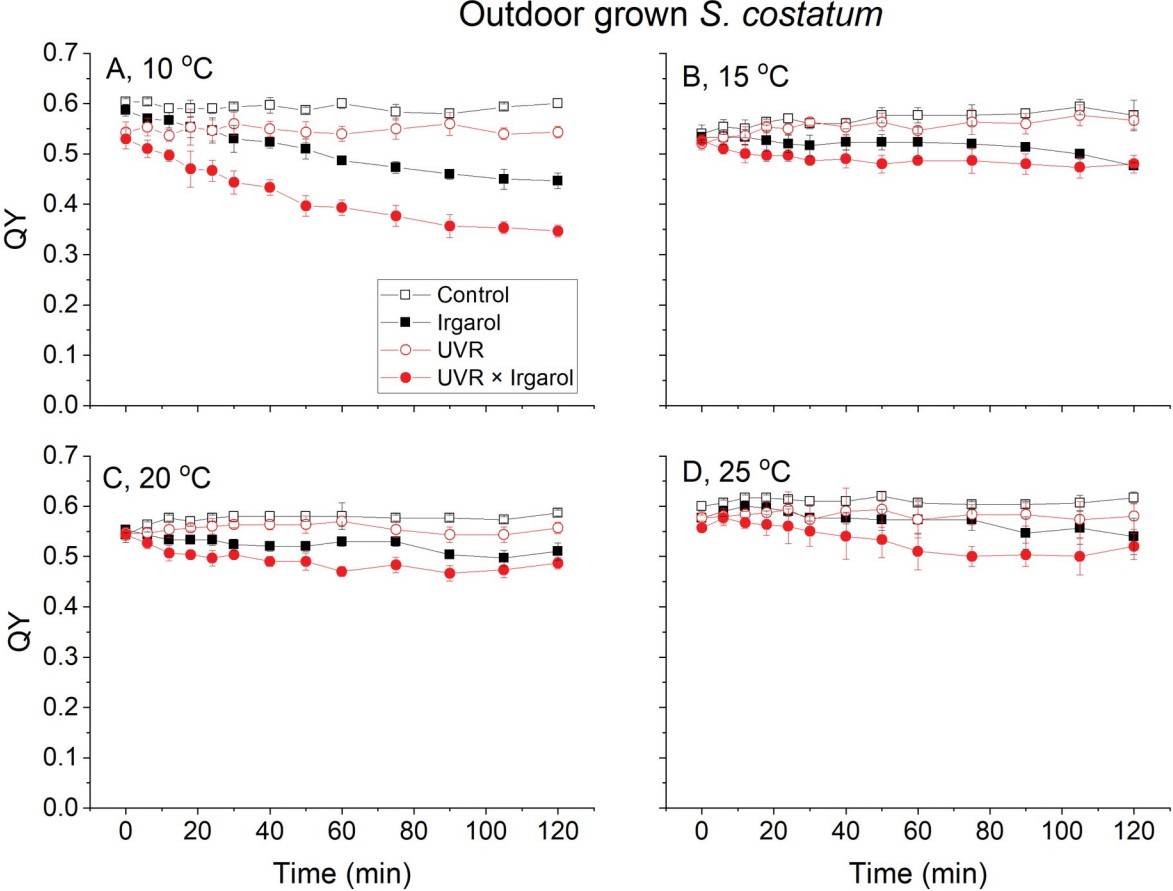

**Fig 3.** The quantum yields of photosystem II of *S. costatum* at four temperatures (A: 10˚C, B: 15˚C, C: 20˚C, D: 25˚C) exposed to two radiation treatments (PAR, black and UVR, red) and two Irgarol 1051 treatments (0, hollow symbols and 0.4 μg L$^{-1}$, solid symbols). Vertical lines represent standard deviations, n = 3.

after a 120-minute exposure under the UVR + Irgarol condition. However, it is worth noting that the decrease in QY amplitude in *S. costatum* was considerably smaller compared to *T. weissflogii* at temperatures ranging from 15˚C to 25˚C, even when cells were exposed to the UVR + Irgarol treatment, with the lowest QY value recorded for *S. costatum* after 120 minutes of exposure, around 0.45 at 15˚C (Fig 3).

The trend of QY for *Entomoneis* sp. exhibited similarities to that of *S. costatum* and *T. weissflogii*, although the initial value of *Entomoneis* sp. was lower at approximately 0.40 (Fig 4), and less decrease in QY observed under warming compared to that observed at 5˚C (Fig 4A and 4B). Statistical analysis shown that the factors applied in this study could independently or interactively affect the photochemistry of diatoms (Table 2). Generally, all factors had significant individual effects on three species, while two-level interactions were observed in temperate species, and a three-level interaction was only found in *T. weissflogii* (Table 2).

To assess the effects of UVR and Irgarol, the relative inhibition was calculated during the period when the QY remained stable after 60 min exposure, and both UVR and Irgarol showed significant inhibition on the tested diatoms. In the case of *T. weissflogii*, the relative UVR inhibition was notably higher at a temperature of 10˚C, reaching approximately 40%. However, as the temperature increased to 15–20˚C, the relative UVR inhibition decreased to around 15%, and further declined to below 10% at a temperature of 25˚C. For *S. costatum*, the relative UVR

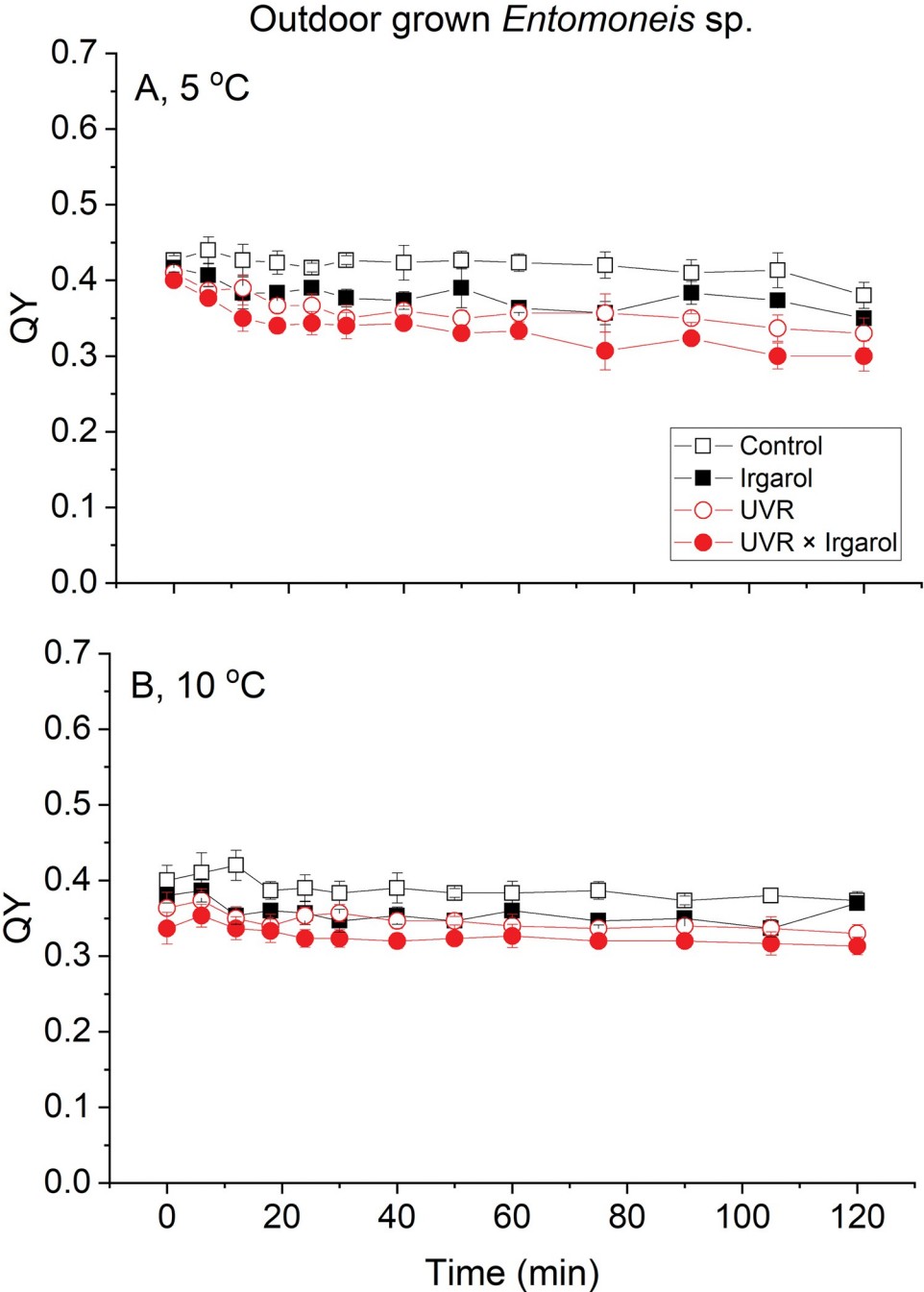

**Fig 4.** The quantum yields of photosystem II of *Entomoneis* sp. at two temperatures (A: 5˚C, B: 10˚C) exposed to two radiation treatments (PAR, black and UVR, red) and two Irgarol 1051 treatments (0, hollow symbols and 0.4 μg L$^{-1}$, solid) symbols. Vertical lines represent standard deviations, n = 3.

inhibition remained below 10% across all temperature levels, with the highest inhibition observed at 10˚C (Fig 5A). Similarly, the relative UVR inhibition on *Entomoneis* sp. was approximately 12% at 5˚C and 8% at 10˚C (Fig 5A). The relative Irgarol inhibition also displayed a similar trend, being highest at the lowest temperature, around 48%, 23% and 10% for *T. weissflogii*, *S. costatum* and *Entomoneis* sp., respectively, and decreasing as the temperature

**Table 2. The statistical results of permutation 3-way ANOVA for the QY of three species during 120 mins exposure experiments under different combinations of UVR, Irgarol and temperature (T), all significant effects were negative.**

| Species | T. weissflogii | | | S. costatum | | | Entomoneis sp. | | |
|---|---|---|---|---|---|---|---|---|---|
| Factor | df | F | Sig. | df | F | Sig. | df | F | Sig. |
| UVR | 1 | 544.331 | **0.000** | 1 | 87.062 | **0.000** | 1 | 142.271 | **0.000** |
| Irgarol | 1 | 3319.491 | **0.000** | 1 | 425.424 | **0.000** | 1 | 56.816 | **0.000** |
| T | 3 | 305.193 | **0.000** | 3 | 43.556 | **0.000** | 1 | 7.931 | **0.012** |
| UVR + Irgarol | 1 | 37.095 | **0.000** | 1 | 7.398 | **0.010** | 1 | 1.544 | 0.232 |
| UVR + T | 3 | 44.117 | **0.000** | 3 | 5.916 | **0.002** | 1 | 6.175 | **0.024** |
| Irgarol + T | 3 | 9.620 | **0.000** | 3 | 23.766 | **0.000** | 1 | 4.288 | 0.055 |
| UVR + Irgarol + T | 3 | 22.524 | **0.000** | 3 | 1.358 | 0.273 | 1 | 0.007 | 0.935 |

increased (Fig 5B). When exposed to simultaneous UVR and Irgarol, the relative U + I inhibition was substantially higher compared to the inhibitions induced by the individual factors, and decreasing with increased temperature (Fig 5C). Interestingly, all relative inhibitions exhibited a correlation with temperature, generally with the lower values observed at the higher temperature.

For treatments that significant interactions were observed, results showed that the ratio of measured QY ($QY_m$) to expected QY ($QY_e$) under UVR + Irgarol were closed to 1.0, varied from 0.99 for *S. costatum*, to 1.08 for *T. weissflogii* (Fig 6). Under the UVR + warming treatments, the $QY_m/QY_e$ values were all greater than 1.0, the lowest values, approximately 1.06, were observed for *S. costatum*, while the highest values, around 1.40, were obtained for *T. weissflogii* (Fig 6). Under Irgarol + warming treatments, the $QY_m/QY_e$ values for both diatoms were greater than 1.0, the highest value, around 1.24, was observed for *T. weissflogii*, while *S. costatum* exhibited a value of approximately 1.16.

## Discussion

In the context of global changes, understanding how phytoplankton respond and adapt is essential for accurately estimating the carbon sink capacity of marine ecosystems [32]. Given the significant variability of seawater temperatures between temperate and polar habitats and their continual rise due to global climate change [33], our study aimed to delve into the possible modulation of the interaction between Irgarol 1051 and UVR caused by temperature increase. The data illustrated that both UVR and Irgarol produced a notable inhibitory impact on the photochemical processes of the three diatoms under examination. However, it's crucial to acknowledge the gap between the current short-term evaluation and the long-term global effects. The observed temperature increase alleviated the short-term inhibitory effect induced by UVR and Irgarol, suggesting that global warming might potentially affect marine ecosystems through its interaction with other environmental stressors. [34].

Marine algae's susceptibility to UV exposure can lead to decreased carbon dioxide assimilation and primary productivity [35]. Our findings are consistent with previous research, which indicated a significant UVR inhibition of the three examined diatom species [36]. The magnitude of UVR inhibition varied among species without a clear correlation to the latitude of their isolation, suggesting that UV sensitivity may evolve over time during extended laboratory storage, regardless of the diatoms' original latitude-light history correlation [37]. Furthermore, when grown at the lowest temperature, *T. weissflogii* showed a significantly higher relative UVR inhibition compared to the other two species. This suggested that the extent of a phytoplankton's sensitivity to environmental stress could correlate with the degree of deviation from its optimal temperature, considering that *T. weissflogii* was isolated from a region warmer than

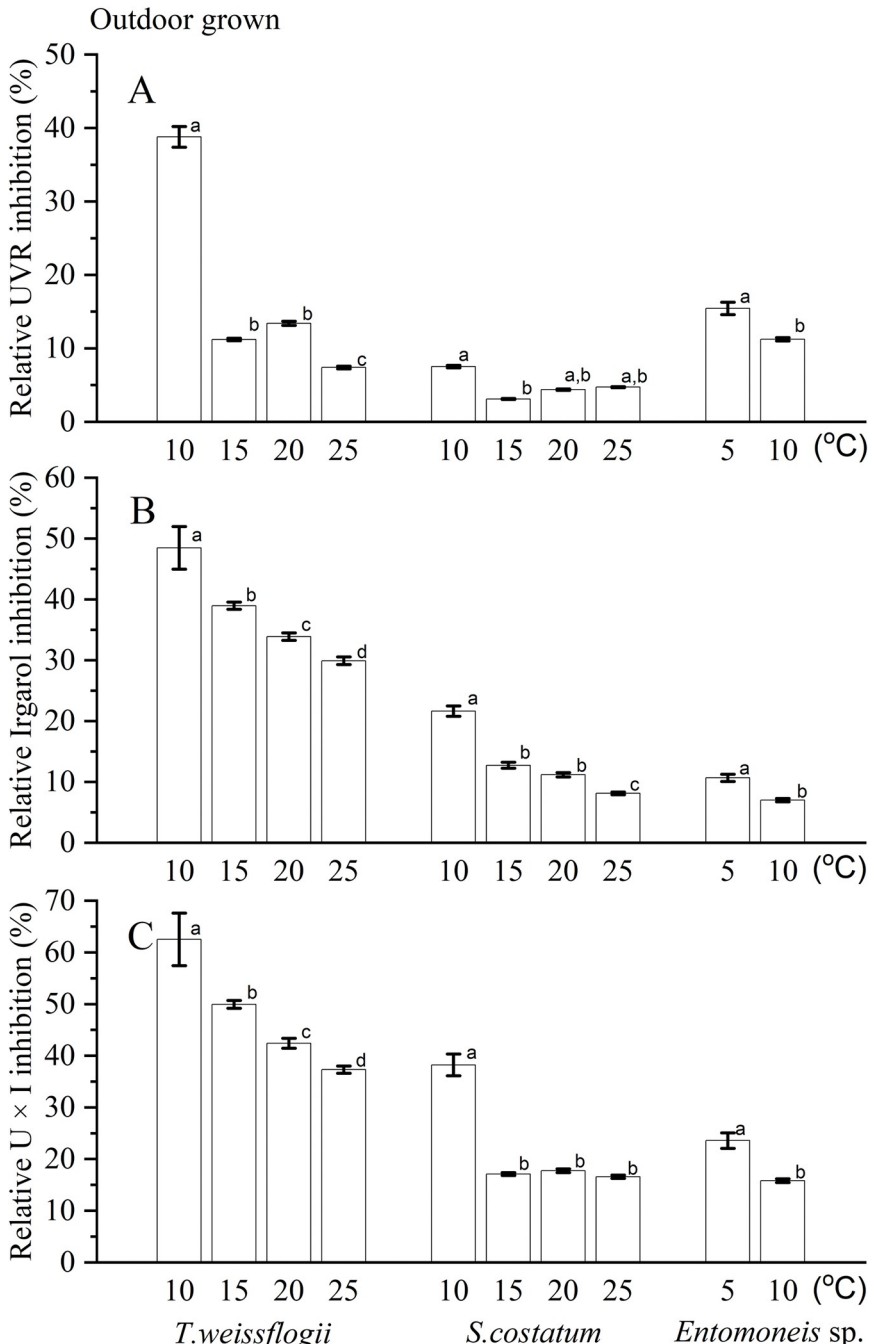

**Fig 5.** Relative UVR inhibition (A), relative Irgarol inhibition (B) and relative U + I inhibition (C) on photosystem II of three diatoms. data are expressed as means ± SD, n = 3, letters above bars indicate significant differences among treatments.

those of the other species, the thermal history could play an important role in response to heat stress, implied that species acclimated at high temperature would be more tolerant and influence diatom species competition [4,38].

As an effective antifouling agent, Irgarol 1051 is more toxic than other triazine compounds [17], that hampers the electron transport on the donor side of PSII by competing for the

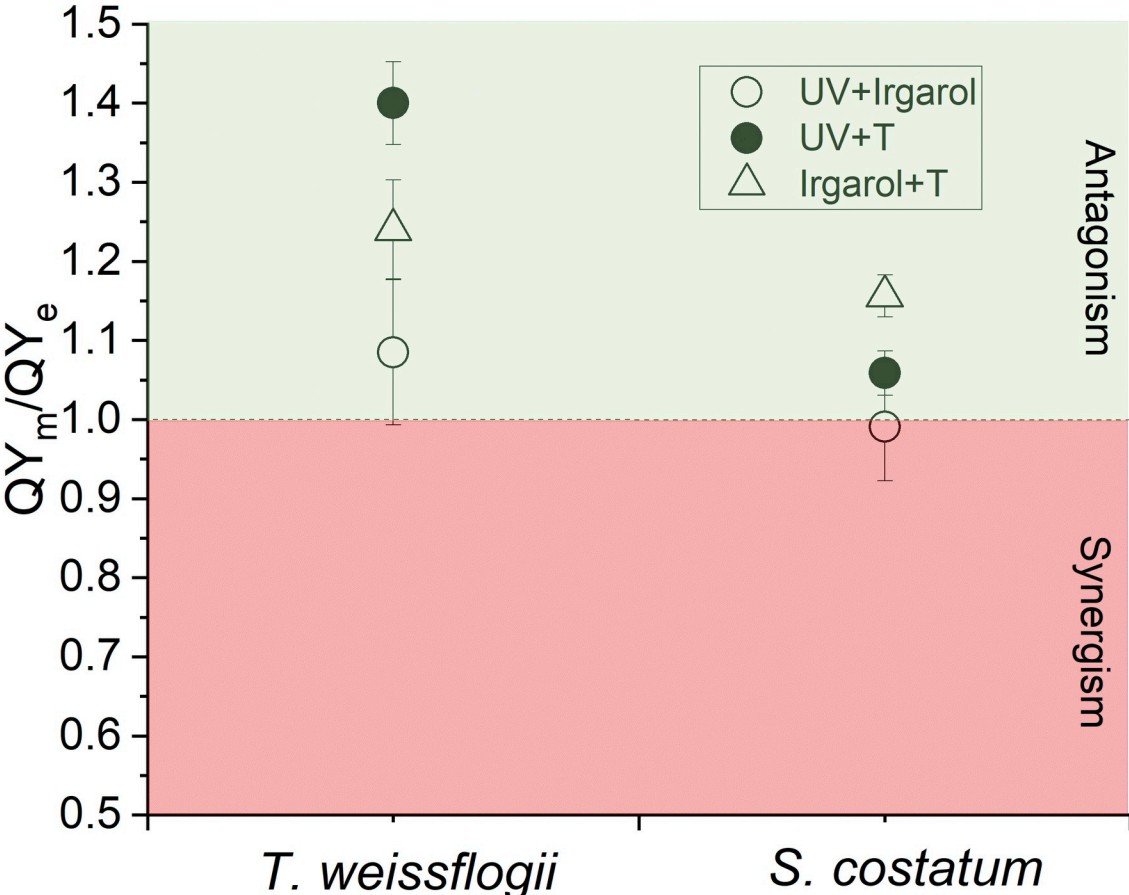

**Fig 6. The ratio of expected QY (QY$_e$) to measured QY (QY$_m$) under UVR + Irgarol, UVR + T and Irgarol + T for *T. weissflogii* and *S. costatum*, vertical lines represent SD, n = 3.**

binding of plastoquininone on the QB protein [16]. If proteins are not adequately replaced, this disruption can lead to a significant inhibition of the D1 protein function and the downstream electron flow [23]. The ambient concentration of Irgarol in certain regions can be relatively high [39], for instance, the accumulated concentration of triazine compounds in the Bohai Sea and the Yellow Sea has been reported to reach up to 1.67 µg L$^{-1}$ [40]. Our findings revealed that Irgarol 1051 can induce up to 45% inhibition on the photosynthetic potential of diatoms, even at relatively low concentrations (0.4 µg L$^{-1}$), indicating that the current environmental levels of organic pollutants might pose substantial ecological risks [41]. Moreover, the impacts of Irgarol 1051 on the photochemistry of three diatoms were species-specific. *T. weissflogii* showed 30–35% inhibition in the typical temperature range (20–25˚C) of the isolated niche, while *S. costatum* and *Entomoneis* sp. showed only 10% inhibition at typical temperatures. These risks could partially account for the observed decrease in phytoplankton productivity [3], as well as the shift of community structures [26,42].

As one of the most important abiotic environmental factors, temperature influences almost all metabolic pathways [43], though marine organisms often encounter multiple environmental stresses [44]. Our data showed that elevated temperature reduced the inhibition of Irgarol 1051 and UVR on the photochemical performance of marine diatoms, while the attenuation of herbicide effects by temperature rise was summarized by Gomes and Juneau [45]. Previous studies found that increasing temperature counteracts the negative effect of UV radiation on

growth and photosynthetic efficiency of cyanobacteria and chlorophytes, possibly by enhancing repair rates of PSII [46,47]. Phytoplankton usually exhibit peak resilience to adverse conditions at their optimal growth temperature [48], thus departures from optimal temperature can modulate the stresses caused by other factors, such as ocean acidification, organic pollutants and ultraviolet radiation, while trade-off between herbicide sensitivity and thermal acclimation/adaptation would also be responsible for the outcome of multiple stressors [49]. The beneficial effect of warming could be linked to its influence on the enzymes involved in the antioxidant system, which helps eliminate reactive oxygen species (ROS) generated under stress [50]. In addition, increasing turnover of PSII proteins would also contribute to the observed beneficial effect of warming [51]. As temperature increases, the enzymatic activity of these protective mechanisms significantly improves, potentially alleviating the impact of environmental stressors on phytoplankton [52].

Phytoplankton are exposed to a multitude of environmental stressors, including organic pollutants, ultraviolet radiation, and ocean acidification [53,54]. With continual human-induced changes, these physical and chemical stressors may interact, potentially influencing marine phytoplankton [55]. Nonetheless, predicting the cumulative impact of these factors on primary production is complex due to the intricacy of their interactions and the limited data available [56]. Furthermore, the burgeoning shipping industry coupled with the delayed recovery of ozone holes in the Antarctic and Arctic might intensify the impact on subarctic phytoplankton, such species have been identified as more sensitive than their counterparts from different regions [57]. The current study, while demonstrating short-term positive effects from elevated temperatures, only investigated a limited spectrum of species and environmental pressures. In addition, it is important to note that the Irgarol concentration applied in this study was nominal rather than an actual value, which may make the results incomparable with those of other studies. In light of the grave challenges phytoplankton could encounter under potential climate change scenarios, it becomes crucial to evaluate the interplay of significant factors, particularly within susceptible ecosystems [58].

## Supporting information

**S1 Dataset. Data of Figs 2–6 was uploaded as Excel file in supporting information.** (XLSX)

## Author Contributions

**Conceptualization:** Yaping Wu.

**Data curation:** Dongquan Bi, Lixin Cao, Yuheng An.

**Formal analysis:** Dongquan Bi, Lixin Cao.

**Funding acquisition:** Yaping Wu.

**Supervision:** Yaping Wu.

**Writing – original draft:** Dongquan Bi, Lixin Cao.

**Writing – review & editing:** Juntian Xu, Yaping Wu.

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
