## [Decision Letter · Decision Letter 0]

25 Sep 2023

PONE-D-23-24973

Increasing temperature alleviates inhibitory effects of Irgarol 1051 and UVR on diatom photochemical performance across latitudes

PLOS ONE

Dear Dr. Wu,

Thank you for submitting your manuscript to PLOS ONE. After careful consideration, we feel that it has merit but does not fully meet PLOS ONE’s publication criteria as it currently stands. Therefore, we invite you to submit a revised version of the manuscript that addresses all the points raised during the review process.

We look forward to receiving your revised manuscript.

Kind regards,

Jean-François Humbert

Academic Editor

PLOS ONE

Journal Requirements:

   "This research was funded by the Natural Science Foundation of China (41876113) to Y Wu, the Key Project of Natural Science of Jiangsu High School (21KJA170001) to Y Wu, Qinglan Project of Jiangsu Province to Y Wu, and Priority Academic Program Development of Jiangsu Higher Education Institutions to Y Wu."

Additional Editor Comments:

**In particular, I ask you, in agreement with the two reviewers, to be much more careful in the abstract, discussion and conclusion of the paper about the interpretation of yours results. I also ask you to change the title of the paper. Not in any way, you can conclude from your data on what will happen in natural ecosystems at different latitudes in the context of global warming. Indeed, the three strains used in your experiments are from being representative of the three species nor of temperate and cold waters (at least you should have several recently isolated starins for each species before starting this discussion). Furthermore, your data only concern the study of very short-term responses to the two stressses applied knowing that many processes acting on different temporal scales are involved in the dynamics of phytoplakton communities.**

Additional point: You have used 3-way RM ANOVA, but you have worked with triplicates. Permutation 3-way ANOVA would be more appropriate, please change.

Reviewers' comments:

Reviewer's Responses to Questions

**Comments to the Author**

1. Is the manuscript technically sound, and do the data support the conclusions?

Reviewer #1: Partly

Reviewer #2: Partly

2. Has the statistical analysis been performed appropriately and rigorously? 

Reviewer #1: Yes

Reviewer #2: Yes

3. Have the authors made all data underlying the findings in their manuscript fully available?

Reviewer #1: No

Reviewer #2: Yes

4. Is the manuscript presented in an intelligible fashion and written in standard English?

Reviewer #1: Yes

Reviewer #2: Yes

5. Review Comments to the Author

Reviewer #1: The authors have indicated that they can provide data on request, but it is not deposited on an accessible site.

This manuscript deals with a subject involving major issues and where work is still too scarce: the interactions between the effects of contaminants and climatic parameters on primary producers. However, this manuscript needs to be reworked in certain parts by supporting them with the bibliography (example see the review by Gomes and Juneau (2017 - doi: 10.3389/fenvs.2017.00050). Thus, the discussion section deserves to be completed on the different behaviors towards applied stresses and their interactions between the 3 species studied. In the Materials and methods section, some descriptive elements are missing. Examples: information on the ecological traits of the species studied, light conditions during acclimatization, duration of UV and IRG exposure during experimentation after acclimatation, etc.

Below are my detailed comments:

Introduction

L35-38 For information, see the recent paper by Courcoul et al (2022 - DOI: 10.1111/1365-2745.14058), which proposes an approach to the effects of temperature on the photosynthetic activity of freshwater periphyton, taking into account not only the intensity of temperature, but also the duration of exposure (an important dimension in qualifying heatwaves): "A thermal dose approach". In addition, an early study investigated the effects of temperature on the sensitivity of a cyanobacteria to a PSII inhibitor (atrazine) and showed that, at higher temperatures, cyanobacteria (in monoculture) are less sensitive to atrazine, so that they expand within phtoplanktonic communities under conditions of higher temperature and herbicide contamination (Bérard et al., 1999 - doi: 10.1007/s002449900541).

L54-55 For information, IRG induces (at real concentrations in the environment) effects on planktonic and periphytic algal communities through the selection of tolerant taxa and thus the change in community composition, implying an acquisition of tolerance to IRG. IRG is more toxic than another PSII inhibitor such as atrazine on different taxonomic groups, particularly diatoms. (Nyström et al. 2002 - Water Research 36 (2002) 2020-2028; Bérard et al., 2003 - doi:10.1016/S0045-6535(03)00674-X ).

Materials and methods

L85 Apart from the fact that these 3 diatoms come from different climatic environments, they are 3 different species. Do you have any information on their life traits and possible sensitivities to different factors such as climate and pollutants?

L89-90 As the historical development conditions of the organisms can influence their response to stress, you have insisted on temperature conditions, but can you also specify the light conditions (obviously the same for the 3 species) during this pre-acclimation phase? Have you taken any measurements? Are these light conditions comparable to those observed in the environments where these strains were isolated? This seems to me an important point to discuss, given that you are testing the effects of light and UV on algae afterwards.

Fig.1 The text in the figure is written in small type. Specify in the legend the color codes (control/IRG) and the concentration of IRG applied, as well as the duration of exposure to the different stresses in the quartz tubes.

L11-112 Specify the exposure time (120 min?) of the cultures in the quartz tubes and justify this choice of exposure time (very short compared with chronic in situ conditions).

L120-124 Why is this table not in the list of tables?

L125 Final IRG concentration of 400ng/L: specify whether this is the nominal or measured concentration in the tubes. Give references on the effects of this level of contamination. Examples: This concentration already inhibits freshwater phytoplankton and periphyton by almost 50% (Nyström et al., 2002, Bérard et al., 2003). Similarly, in the marine environment, concentrations of 60 to 250 ng/L significantly inhibit phytoplankton (Dahl and Blanck, 1996 - https://doi.org/10.1016/0025-326X(96)84828-4). See also the work of Buma et al (2009 - doi:10.1016/j.seares.2008.11.007 ), who tested IRG on T. weissflogii. These elements could also be used in the discussion section of your manuscript (examples L249-250).

L150 Do you consider 60 min to be the time when the effect of applied stresses no longer varies? Specify this before L192.

Discussion

L230-232 Give references to support your assertion

L233-235 Specify (in the Materials and methods section) how long these diatom strains have been isolated and grown in the laboratory (under what conditions?). See also my comment about light conditions during the acclimation phase.

L237-239 See Courcoul et al (2022 - DOI: 10.1111/1365-2745.14058)

L242-245 For information, IRG is much more toxic than atrazine (another PSII inhibitor). See results and discussion in Dahl and Blanck (1996) and Bérard et al. (2003).

L242-253 Discuss the differences in sensitivity to PSII between the 3 species studied.

L251 Cite references showing ecosystem effects through algae. Starting with work on the selection and destructuring effects of algal communities in relation to exposure to IRG.

L251-253 Here you suggest interactions between herbicide and nutrient effects: that's another subject, you need to clarify or find other references that show significant effects of PSII.

L256-258 Can you cite work on the attenuation by temperature rise of the sensitivity of microalgae and cyanobacteria to PSII inhibitors? (see details in Gomes and Juneau 2017 - doi: 10.3389/fenvs.2017.00050)

Reviewer #2: The manuscript by Bi and colleagues submitted for consideration to Plos One report an experimental study regarding short-term effects of UV radiation and an antifouling compound against marine diatoms with varying temperatures. Several issues can be raised, both substantive and formal, that need to be discussed and addressed before acceptance for publication.

The most critical issue is the absence of analytical control of the experimental irgarol exposure, that is assumed to be equal to 0.4 µg/L. The lack of actual exposure data is emphasized as a recurrent issue in experimental evaluation of herbicide toxicity to phytoplankton (see for example the review by Hanson et al. 2019 https://doi.org/10.1016/j.scitotenv.2019.04.336 for atrazine). Without analytical control of irgarol concentration (at least in solvent solution) repeatability and overall strength of the study is flawed. The authors need to address this problem, either by providing actual concentration data or by discussing this point by any means.

I would also raise a general concern about the main direction and conclusions of the submitted manuscript. Authors emphasize the fact that raising temperature decrease the effect of irgarol exposure on photosynthetic potential as measured using variable fluorescence transients. Their conclusions (and the manuscript title) guide readership towards an erroneous view of reduced hazard from marine water herbicide contamination in the frame of global warming. A cautionary discussion is needed, since the experimental procedure here is based on (very) short-term exposure that does not consider adaptation at an individual time scale (i.e. more or less a day, that is a mean doubling time for actively growing diatoms), neither selection of tolerant variants inside a population (at week to month timescale) or competition among phytoplankton populations inside a community in the realm. One should assume that the results presented here are linked to biochemical and toxic-target interactions at very short timescale and that consequences for diatom populations and communities in the field are far from predictable based on these data.

Specific comments:

Title: mentioning here “across latitudes” is confusing, since this was a lab experiment, and maybe a more explicit statement of the provenance of diatom strains would be better suited.

Abstract: I consider that the sentence “These results suggest that elevated temperatures can mitigate…” needs balance or caution, stating that this is correct only at very short term.

References: The introduction section needs more specific references, that can have a global significance and support the current study. I suggest for example considering review papers such as Boyce et al. 2010 (doi:10.1038/nature09268) and / or Tréguer et al. 2018 (10.1038/s41561-017-0028-x) more suited as a context for the present study.

Minor comments:

Lines 91-92: did the authors controlled the cell densities in each culture? Does the current experimental procedure drive each strain to the same overall growth rate?

Lines 98-99: provide more evidence for the absence of effect of DMSO at 0.1% including interactions with temperature – or provide literature references.

Lines 107-108: did the different temperatures resulted in significant differences in observed growth rates?

Line 110: be more specific on acclimation. I assume that 1) temperatures did not change between pre-culture and short term irgarol exposure conditions; 2) the PAR was equal during the two phases; and 3) that changing from polycarbonate bottles to quartz tubes resulted in an UV exposure during short-term experiment. If so, acclimation is encompassing any “bottle effect” and sudden exposure to UVR?

Lines 115-116: technical characteristics of the light filters (material, cutoff lambda…)

Lines 120-124: the Table could be simplified – PAR control / PAB control / PAR + Irgarol / PAB + Irgarol; in current state redundant and confusing.

Line 126: cited reference is for marina waters, quite different from a “typical” coastal environment. Discuss, amend, or provide more references.

Lines 131-133: Light and UVR measurements are more suited for the 2.2. experimental setup section.

Lines 133-136: what is the rationale for choosing a “no dark-adaptation” procedure? What were the biomass / cell densities of each strain for each pre-incubation condition?

Lines 139-following section: give a short explanation of Qy significance (photosynthetic potential?). Consider verifying that the current experiment is actually providing F measurements, not F’ values? I recommend to cite any paper that explain fully the significance of in vivo chlorophyll fluorescence transients – for example Maxwell & Johnson 2000, or any of the numerous ones by Govindjee.

More generally, the authors would have gained information by using the OJIP feature of the AquaPen device, that is able to highlight the different steps of PSII inhibition by herbicide exposure completing Qy evaluation.

Line 148: “U + I” is not defined – see remark above regarding “table” of conditions, and be consistent throughout the manuscript for clarity. Also, why use Y instead of Qy – or precise line 141 that Qy is further noted as Y.

Line 149-150: What is the rationale for the selection of 60-120 minutes exposure to define an average quantum yield?

Line 156-158: since Qy are ratios, it appears not evident that addition of Qy “fractions” is feasible – one should expect that Fm’, or Fo’ can change independently under exposure. Any comment?

Line 185: is there any explanation for a lower Qy in Entomoneis sp.?

Line 222-223: il appears at a shortcut. Does it suggest that UVR and temperature were prone to degrade irgarol as a compound, and / or that the “possible modulation” is to be considered at the metabolic level, where toxicity is compensated by increasing turnover of PSII proteins?

Lines 225-228: this is al least awkward. The gap between the present short-term assessment and global effects with long term pace is extremely wide.

Line 230: Cullen and Lesser 1991 is missing in the reference list.

Line 231: which previous research reports?

Line 237-241: please consider enhancing clarity of this sentence?

Line 249: photosynthetic inhibition or photosynthetic potential? The former needs quantitative assessment, for example using O2 production or CO2 fixation measurements.

Line 253: not fully convinced by the cited references, especially the one about prairie wetlands. See above the example of Boyce et al. 2010 more suited.

Line 260: precise that these studies were on cyanobacteria and chlorophytes – any differences regarding diatom PSII functioning?

Line 261: I really missed the link between the sentence, the Sal et al. reference and what is the intention of the authors here.

Line 264: avoid citing papers without precision – the first one address mixotrophy, and the second one deserves more consideration, since there is also a possible trade-off between herbicide sensitivity, thermal acclimation / adaptation, that could be discussed in the current report.

Table 1: emphasize (bold character) the significant values? And make coherent condition notations throughout. Precise in Table legend that all effects were negative?

Figure 5: harmonize the y axis for all the panels? This would allow a more direct visual comparison of interactions.

6. PLOS authors have the option to publish the peer review history of their article (what does this mean?). If published, this will include your full peer review and any attached files.

Reviewer #1: No

Reviewer #2: No

---

## [Author Response · Author response to Decision Letter 0]

7 Nov 2023

Reviewer #1: The authors have indicated that they can provide data on request, but it is not deposited on an accessible site.

Response: We have submitted the data as supporting information file.

This manuscript deals with a subject involving major issues and where work is still too scarce: the interactions between the effects of contaminants and climatic parameters on primary producers. However, this manuscript needs to be reworked in certain parts by supporting them with the bibliography (example see the review by Gomes and Juneau (2017 - doi: 10.3389/fenvs.2017.00050). Thus, the discussion section deserves to be completed on the different behaviors towards applied stresses and their interactions between the 3 species studied. In the Materials and methods section, some descriptive elements are missing. Examples: information on the ecological traits of the species studied, light conditions during acclimatization, duration of UV and IRG exposure during experimentation after acclimatation, etc.

Response: We appreciated the comments very much, we have thoroughly revised the discussion section as suggested, and the missing information, in M&M also was provided.

Below are my detailed comments:

Introduction

L35-38 For information, see the recent paper by Courcoul et al (2022 - DOI: 10.1111/1365-2745.14058), which proposes an approach to the effects of temperature on the photosynthetic activity of freshwater periphyton, taking into account not only the intensity of temperature, but also the duration of exposure (an important dimension in qualifying heatwaves): "A thermal dose approach". In addition, an early study investigated the effects of temperature on the sensitivity of a cyanobacteria to a PSII inhibitor (atrazine) and showed that, at higher temperatures, cyanobacteria (in monoculture) are less sensitive to atrazine, so that they expand within phtoplanktonic communities under conditions of higher temperature and herbicide contamination (Bérard et al., 1999 - doi: 10.1007/s002449900541).

Response: Thanks for the literatures provided, we have carefully read the papers, and revised our introduction at line 38-39 and 74-75 accordingly, these two papers also be cited now.

L54-55 For information, IRG induces (at real concentrations in the environment) effects on planktonic and periphytic algal communities through the selection of tolerant taxa and thus the change in community composition, implying an acquisition of tolerance to IRG. IRG is more toxic than another PSII inhibitor such as atrazine on different taxonomic groups, particularly diatoms. (Nyström et al. 2002 - Water Research 36 (2002) 2020-2028; Bérard et al., 2003 - doi:10.1016/S0045-6535(03)00674-X ).

Response: Thanks for the information, we have incorporated this information at line 58-60.

Materials and methods

L85 Apart from the fact that these 3 diatoms come from different climatic environments, they are 3 different species. Do you have any information on their life traits and possible sensitivities to different factors such as climate and pollutants?

Response: Thank you for your comment. We appreciate your input. Regarding the sensitivities of Thalassiosira weissflogii, Skeletonema costatum, and Entomoneis sp. to climate change and pollutants, there are indeed some publications available for Thalassiosira weissflogii and Skeletonema costatum, while there are relatively fewer studies on Entomoneis sp. It is important to note that comparing the sensitivities of these species can be challenging due to variations in the experimental conditions used in different studies. However, this study provides a comparative analysis across species, which can be valuable in understanding the potential impacts of climate change and pollutants on various diatom species. 

L89-90 As the historical development conditions of the organisms can influence their response to stress, you have insisted on temperature conditions, but can you also specify the light conditions (obviously the same for the 3 species) during this pre-acclimation phase? Have you taken any measurements? Are these light conditions comparable to those observed in the environments where these strains were isolated? This seems to me an important point to discuss, given that you are testing the effects of light and UV on algae afterwards.

Response: The reviewer is correct that the light condition would also be a very important background for cells that isolated from deferent locations. The solar radiation was monitored by a light meter, and the mean PAR level was 160 μmol photons m-2 s-1 during pre-acclimation. While for a better comparison among three species, the light level for three species was set to the same, which was un-saturated for all species and would not induce any photo-inhibition.

Fig.1 The text in the figure is written in small type. Specify in the legend the color codes (control/IRG) and the concentration of IRG applied, as well as the duration of exposure to the different stresses in the quartz tubes.

Response: The graph and legend was adjusted accordingly.

L11-112 Specify the exposure time (120 min?) of the cultures in the quartz tubes and justify this choice of exposure time (very short compared with chronic in situ conditions).

Response: The exposure duration was set at 120 minutes, as previous research has demonstrated that the photo-physiology of diatoms typically reaches a stable state after 60 minutes of exposure under stressful conditions. Additional information regarding this rationale has been included in lines 115-117.

L120-124 Why is this table not in the list of tables?

Response: We have included this table in the list of tables, and related information in the MS was updated.

L125 Final IRG concentration of 400ng/L: specify whether this is the nominal or measured concentration in the tubes. Give references on the effects of this level of contamination. Examples: This concentration already inhibits freshwater phytoplankton and periphyton by almost 50% (Nyström et al., 2002, Bérard et al., 2003). Similarly, in the marine environment, concentrations of 60 to 250 ng/L significantly inhibit phytoplankton (Dahl and Blanck, 1996 - https://doi.org/10.1016/0025-326X(96)84828-4). See also the work of Buma et al (2009 - doi:10.1016/j.seares.2008.11.007 ), who tested IRG on T. weissflogii. These elements could also be used in the discussion section of your manuscript (examples L249-250).

Response: The concentration in present study is nominal value, we have revised the statement, thanks for the literatures, we have cited in the discussion section.

L150 Do you consider 60 min to be the time when the effect of applied stresses no longer varies? Specify this before L192.

Response: The photochemistry would typically reach a quasi-steady state after 60 min exposure. We have clarified this at line 159-161.

Discussion

L230-232 Give references to support your assertion

Response: references were added as suggested.

L233-235 Specify (in the Materials and methods section) how long these diatom strains have been isolated and grown in the laboratory (under what conditions?). See also my comment about light conditions during the acclimation phase.

Response: The three diatoms were isolated and grown in the lab for over 2 years under 20 degree for T. weisflogii and S. costatum, and 5 degree for polar species. We have added more details at lines 87-100.

L237-239 See Courcoul et al (2022 - DOI: 10.1111/1365-2745.14058)

Response: Thanks for the reference, we have improved the statements at lines 257-260.

L242-245 For information, IRG is much more toxic than atrazine (another PSII inhibitor). See results and discussion in Dahl and Blanck (1996) and Bérard et al. (2003).

Response: Thanks for the reference, we have re-worded the sentence.

L242-253 Discuss the differences in sensitivity to PSII between the 3 species studied.

Response: We have discussed the differences among 3 species at line 282-288.

L251 Cite references showing ecosystem effects through algae. Starting with work on the selection and destructuring effects of algal communities in relation to exposure to IRG.

Response: We have added a reference that ecosystem effects through algae, and made some changes to show the effects of IRG on community structure.

L251-253 Here you suggest interactions between herbicide and nutrient effects: that's another subject, you need to clarify or find other references that show significant effects of PSII.

Response: We have revised the sentence to be more focused.

L256-258 Can you cite work on the attenuation by temperature rise of the sensitivity of microalgae and cyanobacteria to PSII inhibitors? (see details in Gomes and Juneau 2017 - doi: 10.3389/fenvs.2017.00050)

Response: We have cited this reference at line 286-287.

Reviewer #2: The manuscript by Bi and colleagues submitted for consideration to Plos One report an experimental study regarding short-term effects of UV radiation and an antifouling compound against marine diatoms with varying temperatures. Several issues can be raised, both substantive and formal, that need to be discussed and addressed before acceptance for publication. The most critical issue is the absence of analytical control of the experimental irgarol exposure, that is assumed to be equal to 0.4 µg/L. The lack of actual exposure data is emphasized as a recurrent issue in experimental evaluation of herbicide toxicity to phytoplankton (see for example the review by Hanson et al. 2019 for atrazine). Without analytical control of irgarol concentration (at least in solvent solution) repeatability and overall strength of the study is flawed. The authors need to address this problem, either by providing actual concentration data or by discussing this point by any means.

Response: Thank you for your valuable feedback, we acknowledge the concerns you raised, both substantive and formal, particularly regarding the absence of analytical control of the experimental irgarol exposure. Unfortunately, we did not preserve the samples for repeat analysis, making it impossible to re-analyze the irgarol concentration for this study. However, as suggested by the reviewer, we have included a discussion on this point in the revised manuscript at line 321-323. In this section, we address the limitation of not having actual concentration data and emphasize the potential impact it may have on the interpretation of the results. By openly acknowledging this limitation, we aim to provide transparency and ensure that readers are aware of the constraints of our study.

I would also raise a general concern about the main direction and conclusions of the submitted manuscript. Authors emphasize the fact that raising temperature decrease the effect of irgarol exposure on photosynthetic potential as measured using variable fluorescence transients. Their conclusions (and the manuscript title) guide readership towards an erroneous view of reduced hazard from marine water herbicide contamination in the frame of global warming. A cautionary discussion is needed, since the experimental procedure here is based on (very) short-term exposure that does not consider adaptation at an individual time scale (i.e. more or less a day, that is a mean doubling time for actively growing diatoms), neither selection of tolerant variants inside a population (at week to month timescale) or competition among phytoplankton populations inside a community in the realm. One should assume that the results presented here are linked to biochemical and toxic-target interactions at very short timescale and that consequences for diatom populations and communities in the field are far from predictable based on these data.

Response: Thank you for your insightful comments on the direction and conclusions of the submitted manuscript. We agree that caution should be exercised when interpreting our results. To address these concerns, we have revised the title of this manuscript, and include a cautionary discussion in the revised manuscript. 

Specific comments:

Title: mentioning here “across latitudes” is confusing, since this was a lab experiment, and maybe a more explicit statement of the provenance of diatom strains would be better suited.

Response: We have revised the title, now it reads as “Short-term responses of marine diatoms to Irgarol 1051 and UV radiation: Insights into temperature interactions”

Abstract: I consider that the sentence “These results suggest that elevated temperatures can mitigate…” needs balance or caution, stating that this is correct only at very short term.

Response: We have reworded this sentence accordingly.

References: The introduction section needs more specific references, that can have a global significance and support the current study. I suggest for example considering review papers such as Boyce et al. 2010 (doi:10.1038/nature09268) and / or Tréguer et al. 2018 (10.1038/s41561-017-0028-x) more suited as a context for the present study.

Response: We have cited Boyce et al., 2010 at line 37-38. 

Minor comments:

Lines 91-92: did the authors controlled the cell densities in each culture? Does the current experimental procedure drive each strain to the same overall growth rate?

Response: Yes, the cell densities were controlled by exchange with fresh medium every two days, we have provided more details at line 100-103. The growth rates were species specific, T. weisflogii was around 1.1 d-1, S. costatum around 1.8, and polar species around 0.2-0.5 d-1.

Lines 98-99: provide more evidence for the absence of effect of DMSO at 0.1% including interactions with temperature – or provide literature references.

Response: We conducted an initial toxicity test of 0.1% DMSO at 20°C, and no notable inhibition was observed. In response to the reviewer's comment, we subsequently performed toxicity tests on 0.1% DMSO across a range of temperatures from 5 to 25°C, and once again, no significant effects were detected. We have included this additional information in the Materials and Methods section.

Lines 107-108: did the different temperatures resulted in significant differences in observed growth rates?

Response: The growth rates were found to be influenced by temperature, with the lowest temperature resulting in the lowest growth rate during the acclimation period. While we only conducted short-term experiments, thus we only have growth data for the acclimation phase and do not have data corresponding to the exposure experiment conditions.

Line 110: be more specific on acclimation. I assume that 1) temperatures did not change between pre-culture and short term irgarol exposure conditions; 2) the PAR was equal during the two phases; and 3) that changing from polycarbonate bottles to quartz tubes resulted in an UV exposure during short-term experiment. If so, acclimation is encompassing any “bottle effect” and sudden exposure to UVR?

Response: The PAR intensity was found to be similar for both the acclimation and exposure experiments, and this information has been provided at line 100 and 121. Regarding UV exposure, all samples were acclimated in polycarbonate bottles, which create a different UV environment compared to quartz tubes. The polycarbonate bottles allow for similar UVA radiation but almost no UVB radiation. However, since all cells were acclimated under the same conditions, the UV response should still be comparable. To avoid any confusion, we have added an explanation regarding the transmission properties of the polycarbonate bottle at line 98-101.

Lines 115-116: technical characteristics of the light filters (material, cutoff lambda…)

Response: We have provided the information at line 130-132.

Lines 120-124: the Table could be simplified – PAR control / PAB control / PAR + Irgarol / PAB + Irgarol; in current state redundant and confusing.

Response: Revised as suggested.

Line 126: cited reference is for marina waters, quite different from a “typical” coastal environment. Discuss, amend, or provide more references.

Response: Citation was replaced as suggested.

Lines 131-133: Light and UVR measurements are more suited for the 2.2. experimental setup section.

Response: The light condition has been moved to line 119-120.

Lines 133-136: what is the rationale for choosing a “no dark-adaptation” procedure? What were the biomass / cell densities of each strain for each pre-incubation condition?

Response: The non-dark adaption procedure will eliminate recovery during dark adaptation, thus we can get the instant response of cells to the particular treatments in this study. We have added some words at line 154-155

Lines 139-following section: give a short explanation of Qy significance (photosynthetic potential?). Consider verifying that the current experiment is actually providing F measurements, not F’ values? I recommend to cite any paper that explain fully the significance of in vivo chlorophyll fluorescence transients – for example Maxwell & Johnson 2000, or any of the numerous ones by Govindjee.

Response: QY in present study was light adapted value, thanks for the reminding, the fluorescence (F) should be F’, we have revised thorough the text and cited the reference recommended here.

More generally, the authors would have gained information by using the OJIP feature of the AquaPen device, that is able to highlight the different steps of PSII inhibition by herbicide exposure completing Qy evaluation.

Response: This is a good suggestion, unfortuanately, we did not get the OJIP feature in present study, while we definitely will extend the function of AquaPen in our following study.

Line 148: “U + I” is not defined – see remark above regarding “table” of conditions, and be consistent throughout the manuscript for clarity. Also, why use Y instead of Qy – or precise line 141 that Qy is further noted as Y.

Response: U + I represents UVR + Irgarol, which was defined at line 163, and Y was changed to QY.

Line 149-150: What is the rationale for the selection of 60-120 minutes exposure to define an average quantum yield?

Response: We have added the rationale at line 168-170.

Line 156-158: since Qy are ratios, it appears not evident that addition of Qy “fractions” is feasible – one should expect that Fm’, or Fo’ can change independently under exposure. Any comment?

Response: The reviewer is correct. QY are ratios, and commonly used as a measure of photochemical performance for algae. In this study, we specifically used the QY under light conditions, which are the effective values. However, we also considered additional factors such as UV radiation and the presence of Irgarol. The value of QY changed when these factors were added. If the value decreased, it indicated an inhibitory effect caused by the additional factor, and vice versa. Therefore, we calculated the expected values of QY based on the independent effect of each single factor applied to the algae. We then compared these expected values with the real values under specific conditions.

Line 185: is there any explanation for a lower Qy in Entomoneis sp.?

Response: That might be a species-specific difference. Additionally, the seawater used in the present study could also be a contributing factor. Since Entomoneis sp. was isolated far from the land, the seawater from coastal areas may favor the other two species that are isolated from coastal oceans. 

Line 222-223: it appears at a shortcut. Does it suggest that UVR and temperature were prone to degrade irgarol as a compound, and / or that the “possible modulation” is to be considered at the metabolic level, where toxicity is compensated by increasing turnover of PSII proteins?

Response: The reviewer is correct. Previous studies have shown that light can degrade irgarol (DOI: 10.1016/j.marpolbul.2008.09.009), while the degraded compounds still remain toxic to algae. In our present study, we focused on a short-term experiment to examine the photosynthetic response of three diatoms to UVR and irgarol under a warming scenario. We did not specifically investigate the degradation of irgarol or the possible modulation of metabolic processes. However, it is worth noting that the increasing turnover of PSII proteins could potentially compensate for the effects of UVR and irgarol on the diatoms. We have discussed this point in more detail at line 307-308 of our paper.

Lines 225-228: this is al least awkward. The gap between the present short-term assessment and global effects with long term pace is extremely wide.

Response: We appreciate the reviewer's feedback regarding the disparity between the present short-term assessment and the long-term global effects. We acknowledge that our study's short-term findings may not be directly applicable to long-term predictions. We have adjusted our statements to reflect this understanding.

Line 230: Cullen and Lesser 1991 is missing in the reference list.

Response: Sorry, this reference has been added.

Line 231: which previous research reports?

Response: Reference has been added.

Line 237-241: please consider enhancing clarity of this sentence?

Response: We have revised this statement for clarification. 

Line 249: photosynthetic inhibition or photosynthetic potential? The former needs quantitative assessment, for example using O2 production or CO2 fixation measurements.

Response: It is photosynthetic potential, we have revised the statements accordingly.

Line 253: not fully convinced by the cited references, especially the one about prairie wetlands. See above the example of Boyce et al. 2010 more suited.

Response: We have cited the paper by Boyce et al.

Line 260: precise that these studies were on cyanobacteria and chlorophytes – any differences regarding diatom PSII functioning?

Response: We have revised statement. For PSII, since this is a much more conserved complex during evolution, the function would be similar for all photo-autotrophs.

Line 261: I really missed the link between the sentence, the Sal et al. reference and what is the intention of the authors here.

Response: We have removed the sentence that cited Sal et al;, and revised the following sentences to be more logical.

Line 264: avoid citing papers without precision – the first one address mixotrophy, and the second one deserves more consideration, since there is also a possible trade-off between herbicide sensitivity, thermal acclimation / adaptation, that could be discussed in the current report.

Response: We have removed the irrelevant citation, and discussed more based on the second citation at line 302-304.

Table 1: emphasize (bold character) the significant values? And make coherent condition notations throughout. Precise in Table legend that all effects were negative?

Response: Table was revised as suggested

Figure 5: harmonize the y axis for all the panels? This would allow a more direct visual comparison of interactions.

Response: Fig was adjusted as suggested.

---

## [Decision Letter · Decision Letter 1]

24 Nov 2023

PONE-D-23-24973R1Short-term responses of marine diatoms to Irgarol 1051 and UV radiation: Insights into temperature interactionsPLOS ONE

Dear Dr. Wu,

Thank you for submitting your manuscript to PLOS ONE. After careful consideration, we feel that it has merit but does not fully meet PLOS ONE’s publication criteria as it currently stands. Therefore, we invite you to submit a revised version of the manuscript that addresses the two minor points raised during the review process (1.Addition of some references and 2. Minor change in the title of your paper).

We look forward to receiving your revised manuscript.

Kind regards,

Jean-François Humbert

Academic Editor

PLOS ONE

Journal Requirements:

Reviewers' comments:

Reviewer's Responses to Questions

**Comments to the Author**

1. If the authors have adequately addressed your comments raised in a previous round of review and you feel that this manuscript is now acceptable for publication, you may indicate that here to bypass the “Comments to the Author” section, enter your conflict of interest statement in the “Confidential to Editor” section, and submit your "Accept" recommendation.

Reviewer #1: All comments have been addressed

Reviewer #2: All comments have been addressed

2. Is the manuscript technically sound, and do the data support the conclusions?

Reviewer #1: Yes

Reviewer #2: Yes

3. Has the statistical analysis been performed appropriately and rigorously? 

Reviewer #1: Yes

Reviewer #2: Yes

4. Have the authors made all data underlying the findings in their manuscript fully available?

Reviewer #1: Yes

Reviewer #2: Yes

5. Is the manuscript presented in an intelligible fashion and written in standard English?

Reviewer #1: Yes

Reviewer #2: Yes

6. Review Comments to the Author

Reviewer #1: Authors: there are indeed some publications available for Thalassiosira weissflogii and Skeletonema costatum, while there are relatively fewer studies on Entomoneis sp. It is important to note that comparing the sensitivities of these species can be challenging due to variations in the experimental conditions used in different studies.

My answer: Yes, it's difficult to compare work performed under different conditions, but if there are a lot of studies, similar behaviour can sometimes emerge between them. I think you can at least refer to the publications you're talking about for these species (if only to say that there are a few studies for the first two and fewer data for the last one).

Reviewer #2: I acknowledge the changes, comments and precisions that were included in the revised manuscript, in my opinion making it suitable for acceptance.

I have just one suggestion, to keep an important feature of the presented study - in my opinion - change the title for:

"Short-term responses of temperate and subarctic marine diatoms to Irgarol 1051 and UV radiation: Insights into

temperature interactions"

This would add visibility to this experimental work, but it's optional. Depends on other reviewers comments and Editor advice, no need for further reviewing from my point of view on this precise question.

7. PLOS authors have the option to publish the peer review history of their article (what does this mean?). If published, this will include your full peer review and any attached files.

Reviewer #1: No

Reviewer #2: No

---

## [Author Response · Author response to Decision Letter 1]

26 Nov 2023

Reviewer #1: Authors: there are indeed some publications available for Thalassiosira weissflogii and Skeletonema costatum, while there are relatively fewer studies on Entomoneis sp. It is important to note that comparing the sensitivities of these species can be challenging due to variations in the experimental conditions used in different studies.

My answer: Yes, it's difficult to compare work performed under different conditions, but if there are a lot of studies, similar behaviour can sometimes emerge between them. I think you can at least refer to the publications you're talking about for these species (if only to say that there are a few studies for the first two and fewer data for the last one).

Response: We have added a sentence to clarify this point at line 93-95, and related references have been added.

Reviewer #2: I acknowledge the changes, comments and precisions that were included in the revised manuscript, in my opinion making it suitable for acceptance.

I have just one suggestion, to keep an important feature of the presented study - in my opinion - change the title for: "Short-term responses of temperate and subarctic marine diatoms to Irgarol 1051 and UV radiation: Insights into temperature interactions"

Response: Thank you for the suggestion. We agree with the reviewer and have made the necessary changes. The title has been updated accordingly, and the term "cold-water" has been replaced with "subarctic" throughout the entire text.

---

## [Editor Report · Decision Letter 2]

28 Nov 2023

Short-term responses of temperate and subarctic marine diatoms to Irgarol 1051 and UV radiation: Insights into temperature interactions

PONE-D-23-24973R2

Dear Dr. Wu,

We’re pleased to inform you that your manuscript has been judged scientifically suitable for publication and will be formally accepted for publication once it meets all outstanding technical requirements.

Kind regards,

Jean-François Humbert

Academic Editor

PLOS ONE
---

## [Editor Report · Acceptance letter]

30 Nov 2023

PONE-D-23-24973R2 

Short-term responses of temperate and subarctic marine diatoms to Irgarol 1051 and UV radiation: Insights into temperature interactions 

Dear Dr. Wu:

I'm pleased to inform you that your manuscript has been deemed suitable for publication in PLOS ONE. Congratulations! Your manuscript is now with our production department. 

Kind regards, 

on behalf of

Dr. Jean-François Humbert 

Academic Editor

PLOS ONE